# Adaptive Slot Attention: Object Discovery with Dynamic Slot Number

## Abstract

Object-centric learning (OCL) extracts the representation of objects with slots, offering an exceptional blend of flexibility and interpretability for abstracting low-level perceptual features. A widely adopted method within OCL is slot attention, which utilizes attention mechanisms to iteratively refine slot representations. However, a major drawback of most object-centric models, including slot attention, is their reliance on predefining the number of slots. This not only necessitates prior knowledge of the dataset but also overlooks the inherent variability in the number of objects present. To overcome this fundamental limitation, we present a novel complexity-aware object auto-encoder framework. Within this framework, we introduce an adaptive slot attention mechanism that dynamically determines the optimal number of slots based on the content of the data. This is achieved by proposing a discrete slot sampling module that is responsible for selecting an appropriate number of slots from a candidate list. Furthermore, we introduce a masked slot decoder that suppresses unselected slots during the decoding process. To validate the effectiveness of our framework, we conduct extensive evaluations on object discovery tasks using diverse datasets. The experimental results demonstrate that our framework achieves performance that is comparable to or even surpasses the best-performing fixed-slot models in evaluation. Moreover, our analysis substantiates that our method exhibits the capability to dynamically adapt the slot number according to the complexity of each specific instance. The instance-level adaptability offers potential for further exploration in slot attention research.

## 1 Introduction

Object-centric learning marks a departure from conventional deep learning paradigms, focusing on the extraction of structured scene representations rather than relying solely on global features. These structured representations encompass crucial attributes such as spatial information, color, texture, shape, and size, effectively delineating various regions within a scene. These regions, characterized by distinct yet cohesive properties, can be likened to objects in the human sense. These object-centric representations, often referred to as slots, are organized within a set structure that partitions the global scene information.

Traditionally, object-centric learning adopts unsupervised methods with reconstruction as the primary training objective. This process clusters distributed scene representations into object-centric features, with each cluster associated with a specific slot. Decoding these slots independently or in an auto-regressive manner yields meaningful segmentation masks. This inherent characteristic of object-centric learning has paved the way for its application across diverse tasks, including unsupervised object discovery (Locatello et al., 2020; Greff et al., 2019), segmentation (Zadaianchuk et al., 2022), tracking (Kipf et al., 2021a; Elsayed et al., 2022), and manipulation (Singh et al., 2021a). Among these pioneering algorithms, Slot Attention (Locatello et al., 2020) emerges as the most prominent and widely recognized method in the field.

However, a significant challenge within the realm of slot attention is its reliance on a predefined number of slots, which can prove problematic. On one hand, accurately determining the number of objects in a dataset can be challenging, especially when annotations are absent. On the other hand, datasets often exhibit varying object counts, rendering a fixed, predefined number impractical.

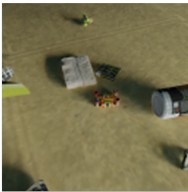 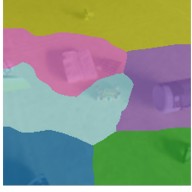 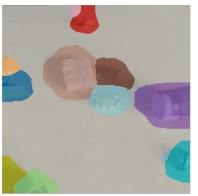 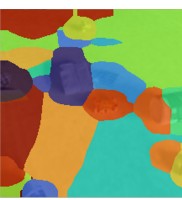

Raw Image     Under-Segmentation     Proper-Segmentation     Over-Segmentation

Figure 1: Illustration of raw image and three kinds of segmentation masks. Pixels colored the same are grouped together as the slot. The numbers of slots are very important.

Incorrectly specifying the number of slots can substantially impact the results, as illustrated in Fig. 1, where an inadequate slot count leads to under-segmentation, while an excessive count results in over-segmentation.

To address this challenge, we present an approach that adaptively determines the number of slots for each instance based on its inherent complexity. Our goal is to allocate a larger slot count for instances with more objects while a smaller number for fewer objects. To achieve this, we propose a novel complexity-aware object auto-encoder framework. Within this framework, we initially generate a relatively large number of slots, denoted as $K_{max}$, and dynamically select a subset of slots for the reconstruction process. Additionally, our framework incorporates a slot sparsity regularization term into the training objective, explicitly considering the complexity of each instance. This regularization term ensures a balance between reconstruction quality and the utilization of an appropriate number of slots.

Our framework encompasses two pivotal strategies to accomplish these objectives. Firstly, we leverage a lightweight slot selection module to acquire a sampling strategy that identifies the most informative slots while discarding redundant ones before reconstruction. This enables us to sample a subset with gradients propagated through the network during training. For seamless end-to-end training, we employ Gumbel-Softmax (Jang et al., 2016) and, for computational efficiency, adopt the mean-field formulation (Blei et al., 2017). Secondly, we introduce a masked slot decoder that adeptly removes information associated with the dropped slots.

We summarize our contributions here: 1) **Novel Framework**: We propose a novel complexity-aware object auto-encoder framework that dynamically determines the number of slots, addressing the limitation of fixed slot counts in object-centric learning. 2)**Efficient Slot Selection**: Our framework incorporates an efficient and differentiable slot selection module, enabling the identification of informative slots while discarding redundant ones before reconstruction. 3)**Effective Slot Decoding**: We present a masked slot decoder that efficiently removes information associated with unused slots. 4)**Promising Results**: Through extensive empirical experiments, we demonstrate the superiority of our approach, achieving competitive or superior results compared to models relying on fixed slot counts. Importantly, our method excels in instance-level slot count selection, showcasing its practical efficacy in various applications.

## 2 RELATED WORK

**Object-Centric Learning**. Object-centric learning fundamentally revolves around the idea that natural scenes can be effectively represented as compositions of distinct objects. Current methodologies in this field mainly fall into two categories: 1) **Spatial-Attention Models** are exemplified by models like AIR (Eslami et al., 2016), SQAIR (Kosiorek et al., 2018), and SPAIR (Crawford & Pineau, 2019). These approaches infer bounding boxes for objects, providing explicit information about an object's position and size. Typically, such methods employ a discrete latent variable $z_{pres}$ to determine the presence of an object and infer the number of objects. However, these box-based priors often lack the flexibility needed to accurately segment objects with widely varying scales and shapes. 2) **Scene-Mixture Models** explain a visual scene by a finite mixture of component images. Methods like MONET (Burgess et al., 2019), IODINE (Greff et al., 2019), and GENESIS (Engelcke et al., 2019) operate within the Variational Autoencoder (VAE) framework. They involve multiple encoding and decoding steps to process an image. In contrast, Slot Attention (Locatello et al., 2020) takes a unique approach by incorporating an iterative procedure within a single encode-decode step.

One significant advantage of slot attention is its ability to generate a set of output vectors (slots) that exhibit permutation invariance. These output slots prove valuable for both unsupervised tasks, such as object discovery, and supervised tasks like set prediction.

Building upon the success of slot attention, various extensions and adaptations have emerged, including SAVi (Kipf et al., 2021b), which extends slot attention to video data using a Transformer decoder, STEVE (Singh et al., 2022), focusing on compositional video generation, and SLATE (Singh et al., 2021b), targeting compositional image generation. However, these methods are often evaluated on synthetic datasets and may exhibit limited performance on real-world data. To bridge this gap, DINOSAUR(Seitzer et al., 2022) proposes an approach that reconstructs deep features in the decoding phase instead of pixel-level reconstruction, demonstrating superior performance on both synthetic and real-world datasets—a method we incorporate in our work.

A common limitation among existing methods in this line is the requirement to predefine the number of slots, often treated as a dataset-dependent hyperparameter. In this context, GENESIS-V2 (Engelcke et al., 2021) introduces a novel approach by clustering pixel embeddings in a differentiable manner using a stochastic stick-breaking process, allowing for the output of a variable number of objects, serving as a valuable baseline method.

**Differentiable Subset Sampling**. Several studies have pursued the goal of achieving differentiable subset selection. Notably, Gumbel-Softmax (Jang et al., 2016; Maddison et al., 2016) introduces a continuous relaxation of the Gumbel-Max trick, enabling the selection of the top-1 element. Building upon this foundation, Gumbel Top-$k$ (Kool et al., 2019) extends the approach to generalize top-$k$ sampling. Another innovative approach, proposed by Xie et al. (2020), approximates top-$k$ sampling by harnessing the Sinkhorn algorithm from Optimal Transport. Furthermore, Cuturi et al. (2019) employs the perturbed maximum method to achieve differentiable selection.

However, a common focus of these works lies in scenarios where the subset size is fixed at $k$, constraining their adaptability for slot number selection. In contrast, our method employs the common mean-field formulation to transform the subset selection problem, which does not rely on a predefined number, into a series of top-1 selections that can be efficiently resolved using Gumbel-Softmax.

## 3 METHOD

**Preliminary**. Slot Attention stands out as one of the most prominent object-centric methods, relying on a competitive attention mechanism. It encompasses both the object-centric representation bottleneck and the entire pipeline. In the pipeline, Slot Attention initially extracts image features using an image encoder $F = f_{enc}(x) \in \mathbb{R}^{H' \times W'}$, where $x$ represents the image. Rather than directly decoding $F$ into $x$, the *Slot Attention Bottleneck* $g_{slot}$ further extracts $K$ slots, denoted as $S_1, \cdots, S_K = \text{SlotAttention}(F)$. The slot attention pipeline proceeds to reconstruct images from these slots using a weighted-average decoder. Each slot $S_i$ is individually decoded through an object decoder $g_{object}$ and a mask decoder $g_{mask}$, subsequently integrated through weighted averaging across the slots.

$$(x_i, \alpha_i) = (g_{object}(S_i), g_{mask}(S_i)), \text{ where } x_i \in \mathbb{R}^{H \times W \times C}, \alpha_i \in \mathbb{R}^{H \times W}. \tag{1}$$

$$\hat{x} = \sum_{k=1}^{K} m_i \odot x_i, \quad m_i = \frac{\exp \alpha_i}{\sum_{l=1}^{K} \exp \alpha_i} \tag{2}$$

We minimize the mean squared error between $x$ and $\hat{x}$ as $\mathcal{L}_{recon}(\hat{x}, x) = \|\hat{x} - x\|_2^2$. Here we utilize a fixed $K$ model as our base model. Moreover, we reconstruct the RGB pixels for toy datasets, while following DINOSAUR to reconstruct feature extracted by self-supervised backbones on more complicated datasets.

### 3.1 COMPLEXITY-AWARE OBJECT AUTO-ENCODER

In the original slot attention approach, a notable limitation is the requirement to predefine the slot number $K$ during training and inference. However, we have observed that the choice of $K$ has a significant impact on the quality of object segmentation results. Smaller values of $K$ fail to adequately separate visual objects, while larger values tend to result in over-segmentation. Intuitively,

the ideal slot number should depend on the actual number of objects present in a given scene, but the number of objects varies across different images in the dataset. To address this issue, we propose a complexity-aware object auto-encoder framework that leverages differentiable sampling methods to dynamically determine the appropriate slot number for each instance.

To address the challenge of slot number selection, we adopt a similar approach as clustering number selection (Blei & Jordan, 2006), where we set an upper bound for the slot number as $K_{max}$. This represents the maximum number of objects an image may contain in the dataset. During the decoding phase, instead of decoding from all slots, our objective is to decode from the most *informative* slots. To achieve this, we learn a sampling method $\pi$ for each instance $\mathbf{x}$. The probability $\pi(z_1, \cdots, z_{K_{max}})$ determines whether to keep or drop each slot $S_{1 \sim K_{max}}$, with $z_i = 0$ indicating the slot $S_i$ should be dropped, and $z_i = 1$ indicating it should be kept during reconstruction. We introduce a masked slot decoder $\hat{x} = f_{dec}(S, Z)$ that effectively suppresses the information of the dropped slots based on $Z$. To further control the slot number we retain, we incorporate a complexity-aware regularization term $\mathcal{L}_{reg}(\pi)$. This regularization term helps ensure the appropriate number of slots are retained based on the complexity of instances. The training objective can be formulated as:

$$
\begin{aligned}
\min \quad & \mathbb{E}_Z \, \mathcal{L}_{recon}\left(\hat{x}, x\right) + \lambda \cdot \mathcal{L}_{reg}\left(\pi\right) \\
\text{where} \quad & S_1, \cdots, S_{K_{max}} = g_{slot}\left(f_{enc}(x)\right) \\
& Z \sim \pi\left(z\right), \ \hat{x} = f_{dec}\left(S, Z\right)
\end{aligned}
\tag{3}
$$

Naturally, without any regularization, the model tends to greedily keep all the slots, as more slots generally lead to better reconstruction quality. In contrast, our complexity regularization, as expressed in Eq. 3, compels the model to achieve the reconstruction objective while utilizing as few slots as possible. The parameter $\lambda$ controls the strength of this regularization.

There are two challenges in achieving the complexity-aware object auto-encoder framework. The first is how to do sampling from a discrete distribution while keeping the module differentiable 3.2. The second is how to design mask slot decoder that is able to suppress the dropped slots 3.3.

## 3.2 MEAN-FIELD SLOT SAMPLING WITH GUMBEL SOFTMAX

Formally, given $K$ slots $S$, there are $2^K$ subsets of $S_{sub} \subseteq S$. To sample from these subsets, we can create a bijection between all $S_{sub}$ and $\{1, \cdots, 2^K\}$, which converts the problem of sampling an indeterminate number set to a top-1 selection problem. This approach builds a comprehensive candidates set and considers the relationship between slots. However, the search space would grow exponentially with the increasing of slots number, resulting difficulties memory usage and model optimization. For example, the neural network can be easily get stuck at the local minima. To address this, we use the mean-field formulation in variational inference (Blei et al., 2017), factoring $\pi$ into a product of independent distributions for each slot:

$$
\pi(z_1, \cdots, z_K) = \pi_1(z_1) \cdots \pi_K(z_K).
\tag{4}
$$

Therefore, the problem of selecting from $2^K$ space is reduced to a $K$ binary selection problem. For each $S_i$, we decide drop or keep the slot individually. This mean-field slot selection approach is computational and sampling efficient. Although the relation among slots is ignored in this step, we postulate this relation can be implicitly modelled by the competition mechanism in slot attention.

To be specific, we denote $S \in \mathbb{R}^{K \times D}$. A light weight neural network $h_\theta : \mathbb{R}^D \to \mathbb{R}^2$ is used to predict the keep/drop probability of each slot individually:

$$
\pi = \text{Softmax}(h_\theta(S)) \in \mathbb{R}^{K \times 2},
\tag{5}
$$

where $\pi_{i,0}$ denote the soft probability to drop the $i$-th slot, while $\pi_{i,1}$ denote the soft probability to keep the $i$-th slot. By applying the Gumbel-Softmax with Straight-Through Estimation (Jang et al., 2016) on the probability dimension and take the last column, we get the hard decision slot mask $Z$:

$$
Z = \text{GumbelSoftmax}(\pi)_{:,1}.
\tag{6}
$$

Here, the colon (:) denotes all rows, and 1 denotes the specific column we want to extract. Since Gumbel Softmax generate onehot vector, take the column we get $K$-dimensional zero-one mask $Z = (Z_1, \cdots, Z_k) \in \{0, 1\}^K$.

### 3.3 MASKED SLOT DECODER

As mentioned in Seitzer et al. (2022), the Transformer decoder is biased towards grouping semantically related instances together, while the mixture decoder is able to separate instances better. The behavior of the mixture-decoder makes it a better choice for exploring dynamic slots since we expect the model to distinguish instances rather than semantics. In this paper, we focus on mixture decoder. With the slots representations $S$ and the keep decision vector $Z$, we introduce several possible design choices of suppressing less important slots based on $Z$.

**Zero slot strategy** directly multiply the zero-one keep decision vector $Z$ with the slots $S$:

$$\tilde{S}_i = Z_i S_i, \tag{7}$$

which shrinks the dropped slots to zero while keeps the others as they are.

**Learnable slot strategy** employs a shared learnable embedding $S_{mask}$ as the prototype of the dropped slot. The intuition is that a learnable dropped slot would offer the model more flexibility and stabilize training, and complement the information loss caused by dropping slots. This is achieved as:

$$\tilde{S}_i = Z_i S_i + (1 - Z_i) S_{mask}. \tag{8}$$

We empirically found that both the two strategies would hurt the reconstruction quality as well as the object grouping. The root cause is that when computing the alpha mask, the zero/learnable-shrinked slots are still decoded to non-zero masks which matter at the softmax operation as follows:

$$m_i = \frac{\exp \alpha_i(\tilde{S}_i)}{\sum_{l=1}^{K} \exp \alpha_l(\tilde{S}_l)}. \tag{9}$$

Therefore, instead of manipulating the slots representations, we propose to shrink the corresponding alpha masks to zero:

$$\tilde{m}_i = \frac{Z_i m_i}{\sum_{l=1}^{K} Z_l m_l + \delta}, \quad m_i = \frac{\exp \alpha_i(S_i)}{\sum_{l=1}^{K} \exp \alpha_l(S_l)}, \tag{10}$$

where $\delta$ is a small positive value for computation stability. We name this strategy as **zero mask strategy**. It is worth noting that neglecting $\delta$, Eq. 10 is equivalent to omitting the slot in the mixture decoder, except that Gumbel-Softmax is applied to ensure differentiability. The key difference is that this strategy manipulates the alpha mask directly, fully removes the information of dropped slot while the other two approaches could not.

For the complexity-aware regularization, we choose to punish the expectation of keeping slots:

$$\mathcal{L}_{reg} = \mathbb{E}\left[\sum_{i=1}^{K} Z_i\right] = \sum_{i=1}^{K} \mathbb{E}\left[Z_i\right]. \tag{11}$$

The smaller expectation, the fewer slot left after selection.

## 4 EXPERIMENTS

**Datasets**. To evaluate its performance, we utilize the challenging MOVi dataset collection(Greff et al., 2022). Specifically, we focus on the MOVi-C and MOVi-E variants, which feature high-quality objects in realistic backgrounds. MOVi-C has up to 10 objects, while MOVi-E includes at most 23 objects. We treat the video-based MOVi datasets as image datasets by flattening the videos. Additionally, we use CLEVR10 (Kabra et al., 2019) as the toy dataset and MS COCO 2017 dataset (Lin et al., 2014) as a real-world dataset, which introduces increased complexity compared to MOVi-C/E.

**Metrics** We use pair-counting, matching-based, and information-theoretic methods for evaluation. The *pair-counting metric* utilizes a pair confusion matrix to compute precision, recall, $F_1$ score, and Adjusted Rand Index. In the *matching-based metric*, we utilize three methods: mBO, CorLoc, and Purity. Purity assigns clusters to the most frequent class, and compute the accuracy of this assignment. mBO calculates the mean intersection-over-union for matched predicted and ground truth

Table 1: Results on MOVi-C. (P., R. for Precision, and Recall).

| Model | $K$ | Pair-Counting | | | | Matching | | | Information | |
| | | ARI | P. | R. | F1 | mBO | CorLoc | Purity | AMI | NMI |
|---|---|---|---|---|---|---|---|---|---|---|
| GENESIS-V2 | 6 | 39.65 | 71.02 | 52.34 | 58.23 | 11.58 | 1.29 | 59.83 | 52.56 | 52.70 |
| | 11 | 26.63 | 65.36 | 37.61 | 45.72 | 14.44 | 6.97 | 49.58 | 40.16 | 40.42 |
| DINOSAUR | 3 | 42.98 | 61.42 | 79.06 | 66.87 | 10.75 | 4.94 | 67.88 | 49.53 | 49.61 |
| | 6 | *73.23* | 83.06 | *84.98* | *82.56* | 33.85 | *73.86* | *83.19* | *76.44* | *76.51* |
| | 9 | 69.11 | *87.50* | 75.53 | 79.08 | *35.00* | 71.26 | 79.77 | 75.43 | 75.50 |
| | 11 | 66.42 | **88.42** | 71.31 | 76.73 | 34.72 | 68.69 | 77.43 | 74.31 | 74.39 |
| Ours | | **75.59** | 84.64 | **86.67** | **84.25** | **35.64** | **76.80** | **85.21** | **78.54** | **78.60** |

Figure 2: Visualization of per-slot segmentation, comparing the fixed 11-slot model(first row) and our model(second row). Dropped slot are left empty.

masks, while CorLoc measures the fraction of images with at least one object correctly localized. The *information-theoretic metric* employs Normalized Mutual Information (NMI) and Adjusted Mutual Information (AMI). Noting that we utilize COCO's instance mask instead of semantic mask. All metrics, except mBO and CorLoc, are computed on the foreground objects. *We use ARI to denote FG-ARI for simplicity.*

**Implementation Details** We employ DINO ViT/B-16 as a frozen feature extractor. We set values of $K_{max}$ to 24 for MOVi-E, 11 for MOVi-C, and 33 for MS COCO 2017. A two-layer MLP is used for each slot to determine the keeping probability. Feature reconstruction is performed using MLP mixture decoder as DINOSAUR. We use Adam optimizer, learning rate $4e-4$, 10k step linear warmup, and exponential learning rate decay. We train 500k steps for main experiments and 200k steps for ablation. Results are averaged over 3 random seeds. More details are in Appendix. We set $\lambda$ to 0.1 for MOVi-E/C and 0.5 for COCO, without specifying a particular claim.

### 4.1 MAIN RESULTS ON EACH DATASET

**Toy Dataset**. We compare a fixed 11-slot model ($K_{max} = 11$) on the toy dataset CLEVR10 in Fig. 2, with pixel reconstruction. The ordinary 11-slot model lacks knowledge of the object number and tends to allocate slots for segmenting the background, resulting in slot duplication. In contrast, our model accurately groups pixels according to the actual number of ground truth objects. Surprisingly, our model exhibits the ability to determine the object count and resolve slot duplication on the toy dataset. Please refer to the appendix for detailed results.

**Results on MOVi-C/E.** Compared to our model, vanilla slot attention in DINOSAUR use a predefined fixed slot number. The selection of slot numbers is subject to the dataset statistics. Note that for data in the wild, we don't have access to the ground-truth statistics. Here, we access the number only for comparison. We established baselines for the MOVi-E dataset with an average of 12 objects (max 23) using small (3, 6, 9), medium (13), and large (18, 21, 24) slot numbers. For the MOVi-C dataset with a maximum of 10 objects, we used slot numbers 3, 6, 9, and 11. Besides, GENESIS-V2 is compared. The results are displayed in Tab. 1, Tab. 2 and Fig. 5.

For *Object Grouping*, our algorithm demonstrates its benefits through three different kinds of metrics. Our method outperform GENESIS-V2 by a large margin. When compared to the fixed-slot DINOSAUR, our complexity-aware model achieves the highest ARI and $F_1$ score, indicating that it can effectively group sample pairs within the same cluster as defined by the ground truth. In terms of Purity, our model yields the highest results, showing the greatest overlap between our predictions

Table 2: Experiments on MOVi-E. (P., R. for Precision, and Recall)

| Model | $K$ | Pair-Counting | | | | Matching | | | Information | |
|---|---|---|---|---|---|---|---|---|---|---|
| | | ARI | P. | R. | $F_1$ | mBO | CorLoc | Purity | AMI | NMI |
| GENESIS-V2 | 9 | 48.19 | 61.52 | 58.86 | 58.14 | 11.16 | 12.38 | 60.07 | 65.16 | 65.35 |
| | 24 | 34.27 | 62.97 | 34.87 | 43.32 | 16.12 | 21.13 | 48.34 | 57.57 | 58.06 |
| DINOSAUR | 3 | 36.78 | 41.37 | 85.27 | 54.10 | 6.23 | 1.67 | 53.19 | 50.31 | 50.42 |
| | 6 | 68.68 | 68.20 | **88.66** | 75.66 | 12.04 | 27.92 | 73.81 | 76.52 | 76.62 |
| | 9 | *76.01* | 77.29 | *87.83* | *81.16* | 25.41 | 87.45 | *79.57* | 81.17 | 81.28 |
| | 13 | 73.74 | 83.73 | 77.35 | 78.93 | 29.08 | *90.02* | 78.41 | *81.53* | *81.67* |
| | 18 | 68.89 | 86.08 | 68.36 | 74.46 | 29.57 | 86.71 | 74.60 | 80.19 | 80.35 |
| | 21 | 66.15 | *87.15* | 63.87 | 71.86 | *30.01* | 85.57 | 72.39 | 79.33 | 79.51 |
| | 24 | 61.98 | **88.09** | 57.82 | 67.91 | **30.54** | 85.15 | 68.96 | 77.93 | 78.14 |
| Ours | | **76.73** | 85.21 | 80.31 | **81.42** | 29.83 | **91.03** | **81.28** | **83.08** | **83.20** |

Table 3: Experiments on COCO datasets. (P., R. for Precision, and Recall)

| Model | $K$ | Pair-Counting | | | | Matching | | | Information | |
|---|---|---|---|---|---|---|---|---|---|---|
| | | ARI | P. | R. | $F_1$ | mBO | CorLoc | Purity | AMI | NMI |
| GENESIS-V2 | 6 | 25.39 | 58.95 | 40.49 | 44.60 | 15.42 | 7.77 | 52.39 | 33.55 | 34.15 |
| | 33 | 9.74 | 63.61 | 10.77 | 15.28 | 10.19 | 0.41 | 21.26 | 24.08 | 26.08 |
| DINOSAUR | 4 | 30.85 | 75.95 | 61.93 | 62.86 | 17.75 | 17.95 | 61.09 | 37.30 | 37.35 |
| | 6 | **41.89** | 82.00 | **70.12** | **70.66** | *27.46* | **50.81** | **69.07** | **46.11** | **46.16** |
| | 7 | *39.95* | 82.87 | 65.69 | 68.00 | **27.77** | *50.09* | 66.40 | *45.25* | *45.31* |
| | 8 | 37.60 | 83.83 | 59.86 | 64.38 | 26.93 | 45.68 | 62.93 | 44.36 | 44.43 |
| | 10 | 35.25 | 85.29 | 54.05 | 60.43 | 27.19 | 44.18 | 59.15 | 43.66 | 43.73 |
| | 12 | 32.70 | 86.44 | 48.63 | 56.53 | 27.02 | 42.42 | 55.55 | 42.64 | 42.71 |
| | 20 | 26.55 | *88.93* | 36.31 | 46.00 | 25.43 | 35.28 | 46.18 | 40.00 | 40.10 |
| | 33 | 20.83 | **90.96** | 26.63 | 36.50 | 24.09 | 32.09 | 37.87 | 37.10 | 37.23 |
| Ours | | 39.00 | 81.86 | *66.42* | *68.37* | 27.36 | 47.76 | *67.28* | 44.11 | 44.17 |

and the foreground in the ground truth. Additionally, the information-based metrics AMI and NMI indicate that our model shares the most amount of information with the ground truth. Overall, our model outperforms fixed slot models across all five mentioned metrics. For *Localization*, our model have the highest CorLoc and as good as best mBO compared with fixed slot models. Improper slot number will oversegment or undersegment the objects, and decrease the IoU, leading to poor spatial localization.

In MOVi-E, 18-24 slots model keeps the precision at a higher level. Our model can decide the slot number according to the instance and further merge the oversegmented clusters together to improve the recall rate by a large amount. On MOVi-E, our model keeps the same level precision with 18-slot model but have around 12 points higher recall. Therefore, our model reach best $F_1$ and ARI scores.

**Results on COCO**. MS COCO has a problem of extreme imbalance in its validation set: most of images has less than 10 objects. This makes it difficult to determine the correct number of slots. To address this, we conducted experiments using a wide range of slot numbers with non-uniform spacing. The results can be found in Table 3 and Fig 5.

When it comes to object grouping, MS COCO is highly sensitive to the number of slots in the fixed-slot DINOSAUR. The experiment showed that the best results were achieved with 6 slots. However, increasing the number of slots led to a rapid decline in performance, especially in object grouping. For example, just going from 6 to 8 slots resulted in a significant drop of around 4 points in ARI, which is about a 10% reduction from the maximum score.

Our models, set $K_{max} = 33$ and equipped with complexity-aware regularization, effectively surpass the performance of the 33-slot model. Specifically, our model achieves approximately 20 points

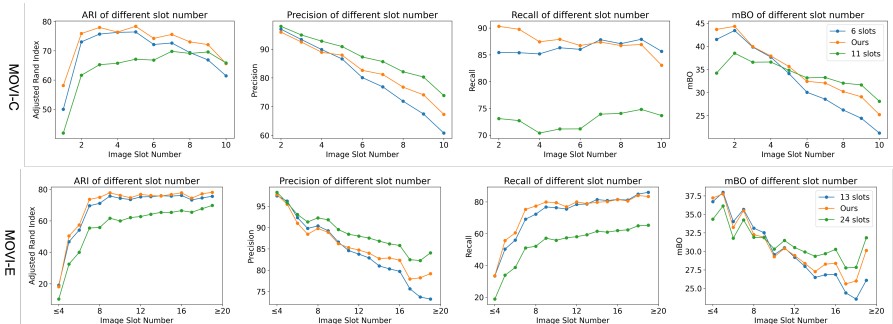

Figure 3: Stratified statistics of four metrics of our models and two fixed slot models, one set the slot number to the upper bound and another set to slot with both high ARI and mBO. We apply stratified sampling according to ground truth object number the image have. The first row is MOVi-C while second row is MOVi-E.

higher in terms of ARI. Although the improvement in localization is comparatively smaller, our model still outperforms the 33-slot model by three points in terms of mBO.

It is worth noting that on the MS COCO dataset, the best results obtained with fixed slot numbers are marginally superior to our results. COCO's nature images present greater challenges than MOVi-C/E due to incomplete labeling, cluttered compositions without clear backgrounds, and a vast range of object sizes and varieties. Despite these challenges, our complexity-aware module enables our model to achieve results comparable to top-performing fixed-slot methods, highlighting its effectiveness.

## 4.2 REVEALING THE INSIGHTS OF OUR MODEL

**Statistical Results Stratified by Ground-truth Object Number**. The above sections reflect the average performance of models on the whole validation datasets. However, the model may over-fit a specific slot number to improve the final average. To eliminate this possibility, we used stratified sampling method on MOVi-C/E to display the values of various metrics of images with different ground truth object number in Fig. 3 . For MOVi-C, we compare our models with fixed 11 slots(the upper bound of object number) and fixed 6 slots(high ARI and mBO simultaneously). Similarly, for MOVi-E, compare our models with fixed 13-slot and 24-slot models.

*Precision&Recall* are inversely related to the number of objects present in an image. As the number of objects increases, precision decreases while recall increases. In the case of our model, it falls somewhere in between high-slot and low-slot models in terms of precision. However, regarding the recall, our model outperforms high-slot models significantly and performs just as well as low-slot models for image with different objects number.

*ARI&mBO*. Different advantages can be observed for large and small slot models. Our model's curve encompasses the metric curve of the two fixed-slot models for ARI, indicating a wider range of effectiveness. For mBO, our model achieves a performance comparable to the better-performing fixed-slot models across the entire range. This demonstrates the efficacy of our dynamic slot selection approach, as it consistently delivers favorable results.

**Analyzing the Slot Selection Process**. We reveal the insights of our model by showing some examples in Fig. 5, and heatmap and slot distribution in Fig. 4. The predictions of fixed-slot models tend to be concentrated within a narrow range, forming a sharp peak which deviates from ground truth distribution. In contrast, our models exhibit a smoother prediction distribution that closely aligns with the ground truth.

On MOVi-C/E, fixed-slot models may generate fewer masks due to the one-hot operation. However, most of their predictions are concentrated around the predefined slot number, resulting in a heatmap exhibiting a distinct vertical pattern. Our model instead exhibits an approximately diagonal pattern on the heatmap. In other words, our model can predict more masks for images with more objects, and the number of predicted masks roughly matches the ground truth number. Though the diagonal relationship is imperfect, and the prediction on images with an extremely large or small number of objects is slightly poorer than other images, our model first achieves the adaptive slot selection.

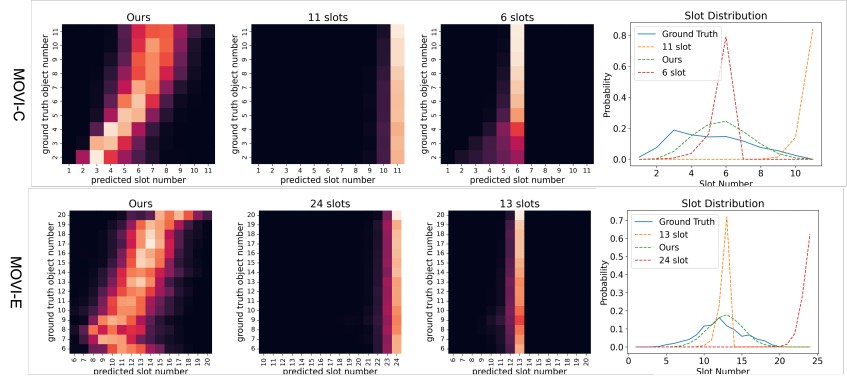

Figure 4: Heatmap of confusion matrix and slot distribution of our models and two fixed slot models on MOVi-C/E. For heatmap, $y$-axis corresponds to the number of objects of ground truth, and $x$-axis is the predicted object number by models. Due to imbalanced ground truth object numbers, we normalized the row and visualize the percentage. The brighter the grid, the higher the percentage. The slot distribution graph shows the probability density of grounded and predicted object numbers.

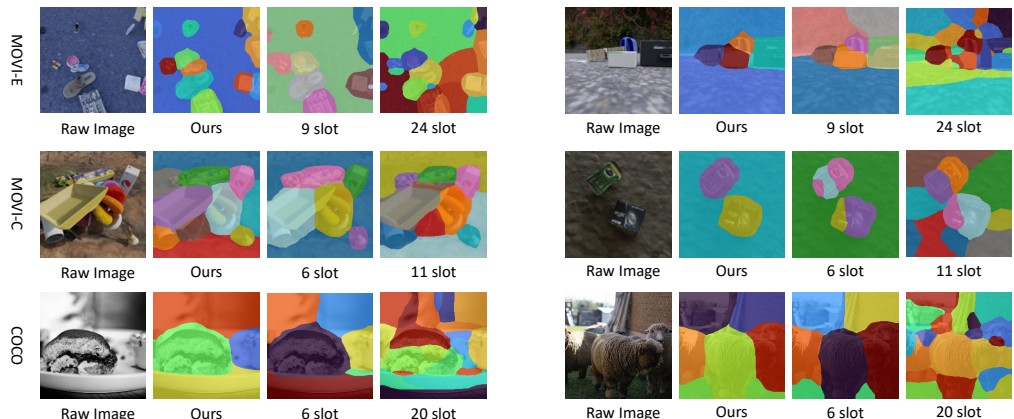

Figure 5: Visualization of our models and the fixed-slot DINOSAUR on three datasets. For each dataset, we select two examples and compare our model with a small slot number and a large slot number.

Figure 5 demonstrated the adaptability of slot numbers at the instance level with illustrative examples. In particular, on the MOVi-E dataset, our model successfully generates 13 and 6 slots for two different images, highlighting a significant discrepancy in slot counts. Noteworthy, our results effectively group pixels based on image complexity, resulting in accurate and appropriate segmentation.

**More Ablation Study in Appendix**. We also conduct extensive ablation to evaluate each component of our framework in the appendix. Particularly, we make the comparison of three designs of masked decoder. Further, we demonstrate the necessity of Gumbel Softmax, and the influence of $\lambda$. Extensive ablation show the efficacy of our model.

## 5 CONCLUSION

We have introduced a complexity-aware object auto-encoder framework that is able to dynamically determine appropriate slot number according to the content of the data in object-centric learning. The framework composes of two parts. A slot selection module is first proposed based on Gumbel-Softmax for differentiable training and mean-field formulation for efficient sampling. Then, a masked slot decoder is further designed to suppress the information of unselected slots in the decoding phase. Extensive studies demonstrate the effectiveness of our framework in two folds. First, our framework achieves comparable or superior performance to those best-performing fixed-slot models. Second, our framework is capable of selecting appropriate slot number based on the complexity of the specific image. The instance-level adaptability offers potential for further exploration in slot attention.

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

## 6 APPENDIX

### 6.1 MORE IMPLEMENTATION DETAILS FOR MOVI-C/E AND COCO

**Vision backbones** We utilize the Vision Transformer backbone and leverage the pre-trained DINO weights available in the timm Wightman (2019) library. Our specific configuration entails using ViT-B/16, which consists of 12 Transformer blocks. These blocks have a token dimensionality of 768, with a head number of 12 and a patch size of 16. In our pipeline, we take the output of the final block as the input of slot attention module and the reconstruction target.

**Slot Attention** We adopt the slot attention bottleneck methodology based on the original work (Locatello et al., 2020) for our implementation. The slot initialization process involves sampling from a shared learnable normal distribution $\mathcal{N}(\mu, \Sigma)$. Throughout all the experiments, we iterate the slot attention mechanism with 3 steps. The slot dimension is set to 128 for MOVi-C/E and 256 for COCO datasets. For the feedforward network in Slot Attention, we utilize a two-layer MLP (Multi-Layer Perceptron). The hidden dimension of this MLP is set to 4 times the slot dimension.

**Light Weight Network for Probability Prediction** We utilize a two-layer MLP for the probability prediction. The hidden dimension of this MLP is set to 4 times the slot dimension, and the output dimension is set to 2.

**Decoder** We utilize a four-layer MLP with ReLU activations in our approach. The output dimensionality of the MLP is $D_{feat} + 1$, where $D_{feat}$ represents the dimension of the feature, and the last dimension is specifically allocated for the alpha mask. The MLP's hidden layer sizes differ based on the dataset used. For the MOVi datasets, we employ hidden layer sizes of 1024. On the other hand, for COCO, we utilize hidden layer sizes of 2048.

**Optimizer** In our main experiments, we train the models for 500k steps. While for the ablations, we train them for 200k steps. To optimize the model's parameters, we employ the Adam optimizer with a learning rate of $4e - 4$. The $\beta_0$ and $\beta_1$ parameters are set to their default values $\beta_0 = 0.9, \beta_1 = 0.999$.

To enhance the learning process, we incorporate a learning rate decay schedule with linear learning rate warm-up of 10k steps. The learning rate follows an exponentially decaying pattern, with a decay half-life of 100k steps. Furthermore, we apply gradient norm clipping, limiting it to a maximum of 1.0, which aids in stabilizing the training procedure.

The training of the models takes place on 8 NVIDIA T4 GPUs, with a local batch size of 8.

### 6.2 MORE IMPLEMENTATION DETAILS FOR CLEVR10

For the experiment on toy dataset CLEVR10, we do pixel-level reconstruction instead of feature reconstruction. We utilized the CNN feature encoder and boardcast decoder in (Locatello et al., 2020). We set the slot dimension to 64, and set the hidden dimension to 128 for the feed-forward network in slot attention. The other setting closely follow the experiments on MOVi-C/E and COCO.

### 6.3 DETAILED RESULTS ON TOY DATASET

In Tab. 4, we quantitatively compare our model with several fixed-slot models on the toy dataset CLEVR10 under pixel reconstruction setting. Moreover, we provide qualitative comparison among our model, 6-slot model, and 11-slot model in Fig. 6. The 11-slot model often assign one or more slots to represent the background, while 6-slot model can not properly segment all objects when the image have more than 6 objects.

Our model differs significantly from the 11-slot model in terms of handling the background, as observed from the visualizations. In the case of the 11-slot model, when the number of objects of an image is small, the 11-slot model tends to divide the background into several slots. However, this division does not segment the background into several regions. Instead, the segmentation of background is very even in terms of light and shadow.

On the contrary, our model takes a different approach of not utilizing a fixed background slot. Instead, it intelligently merge the background regions to the nearest foreground objects. It is reflected

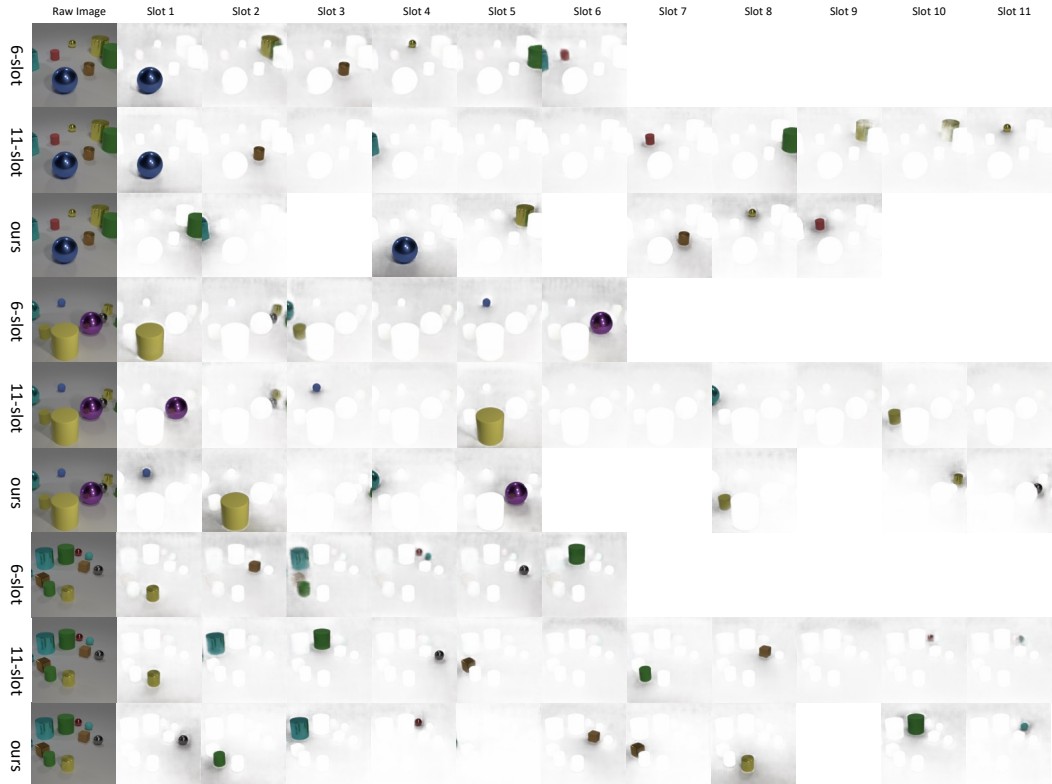

Figure 6: Illustration of per-slot segmentation of our model and two fixed-slot models on CLEVR10.

Table 4: Results on CLEVR10 dataset.

| Models | Pair-Counting | | | | Matching | | | Information | |
|---|---|---|---|---|---|---|---|---|---|
| | ARI | P. | R. | $F_1$ | mBO | CorLoc | Purity | AMI | NMI |
| 3 | 59.00 | 60.85 | 93.17 | 72.22 | 10.33 | 0.08 | 70.09 | 66.36 | 66.41 |
| 6 | 90.77 | 89.26 | 98.13 | 93.08 | 19.35 | 19.45 | 91.81 | 92.32 | 92.34 |
| 9 | 97.59 | 97.86 | **98.55** | 98.14 | *26.45* | *45.72* | 97.81 | 97.39 | 97.40 |
| 11 | **98.06** | **98.77** | 98.35 | **98.51** | **27.39** | **47.15** | **98.27** | **97.90** | **97.90** |
| Ours | *97.65* | *98.19* | *98.36* | *98.21* | 22.51 | 37.00 | *98.03* | *97.50* | *97.51* |

in the visualization that the shadow (which corresponds to background) around the object is much darker in our proposed model than the fixed slot model. The visualizations demonstrate that our model consistently outputs an appropriate number of slots for each image. In order to evaluate the accuracy of our model in determining the number of objects, we illustrate the heatmap of confusion matrix of segmentation number and the slot distribution of the models in Fig. 7. Our models exhibit a prediction distribution that almost perfectly aligns with the ground truth. Additionally, the heatmap revealed an excellent diagonal relationship, indicating that our method can roughly resolves the challenge of unsupervised object counting on CLEVR10. The diagonal of the heatmap reveals the instance-level adaptability of our model.

As for the metrics, our model achieves comparable object grouping results to both the 11-slot and 9-slot models. However, when it comes to localization, our model exhibits slightly lower performance. Nonetheless, we would like to suggest that this discrepancy can be attributed to the distinct approach we take in handling the background. Our model tends to merge the shadows around objects with the foreground, which, in turn, results in slightly lower IoU scores for the object masks predicted by our model. Consequently, this leads to drops in metrics such as mBO and CorLoC.

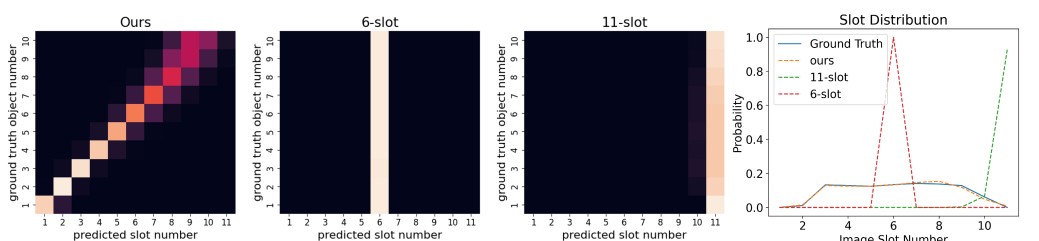

Figure 7: Heatmap of confusion matrix and slot distribution of our models and two fixed slot models on CLEVR10. For heatmap, $y$-axis corresponds to the number of objects of ground truth, and $x$-axis is the predicted object number by models. Due to imbalanced ground truth object numbers, we normalized the row and visualize the percentage. The brighter the grid, the higher the percentage. The slot distribution graph shows the probability density of grounded and predicted object numbers.

## 6.4 MORE ANALYSIS ON COCO

Similarly, we present the heatmap of the confusion matrix of segmentation number and the slot distribution of the models in Figure 8. However, It is worth noting that the COCO dataset has incomplete annotations, which means that not all objects have been annotated. In this case, we make our method solely focus on predictions related to the foreground. In other words, we only consider slots whose masks intersect with foreground objects. Besides, we limit our analysis to images that contain no more than 10 objects, since a significant majority of COCO images contain fewer than 10 objects. These particular images play a crucial role in determining an appropriate value for the fixed slot number, as 6 slot number reached the best results on COCO. Among the three models, our model shows better correlation between the ground truth object number and the predicted slot number. In contrast, the fixed-slot models fail to exhibit this diagonal pattern, further highlighting the efficacy of our approach.

As for the distribution of total slot number, all three models' predictions deviate from the ground truth. However, our model demonstrates the closest approximation to the ground truth distributions. This is substantiated by the visual examples presented in Figure 13, where our model showcases its ability to generate semantically coherent and meaningful segmentations. Notably, our model demonstrates adaptability by adjusting the slot number according to the complexity of the images, thereby further enhancing the quality of its predictions.

Figure 13 provides valuable insights into the reasons behind the deviations of the distributions from the ground truth. Let's consider the last row of Fig. 13, where our model demonstrates successful segmentation of the raw image into distinct regions, including the head of the girl, the T-shirts, the glove, and the background. The separation of the T-shirt and the head seems to be an over-segmentation compared to annotation, which may lead to low metric score. However, each segmented region exhibits semantic coherence and is still visually reasonable.

Real-world datasets often encompass complex part-whole hierarchies within objects. Without the availability of human annotations, accurately segmenting objects into the expected part-whole hierarchy becomes extremely challenging. Since many objects consist of multiple parts, just like the human body, it is expected that our model's predictions will slightly surpass the ground truth in terms of the number of slots. As a result, our model's prediction will be slightly more than ground truth.

## 6.5 ABLATION

We conduct a series of ablation studies on MOVi-E dataset to investigate the components and design choices of our method.

**Comparison of three design choices of masked slot decoder.** In our main paper, we proposed several design choices of the masked slot decoder, and we focused on the zero mask variant. In Tab. 5 and Fig. 9, we compare the three variants in both quantitative and qualitative ways. The results show that our zero mask method effectively improves most metrics compared to the original

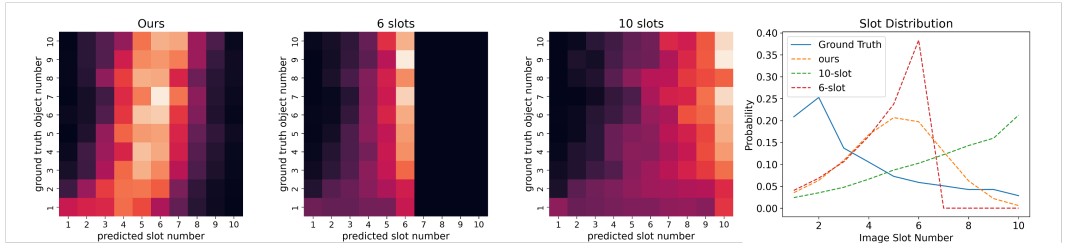

Figure 8: Heatmap of confusion matrix and slot distribution of our models and two fixed slot models on COCO. The sole distinction is that we consider the ground truth masks and predicted masks on the foreground.

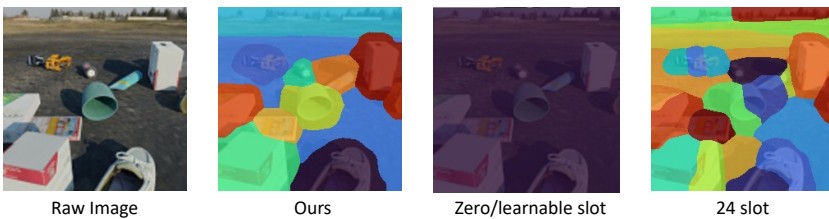

Figure 9: Illustration of the segmentation mask of three designs of mask slot decoders and ordinary 24-slot model.

slot attention model with 24 slots. However, in zero slot and learnable slot strategy, simply changing the manipulation on the mask to the manipulation on the slot makes the model collapse. Both zero slot and learnable slot strategy tend to group all pixels together instead of making a segmentation. If we do not explicitly remove the effect of the dropped slot by setting their alpha masks to zero, the zero/learnable slot will still contribute to the reconstruction. Some instance-irrelated information will be introduced and may mislead the slot selection. As a result, zero/learnable slot tend to group all pixels together.

**The Necessity of Gumbel Softmax.** In the main paper, we utilized the hard zero-one mask:

$$Z = \text{GumbelSoftmax}(\pi)_{:,1}. \tag{12}$$

To verify the necessity of Gumbel-Softmax, we provide experiments that keep the same masked slot decoder but replace the hard mask with a soft mask without Gumbel Softmax:

$$Z_{soft} = \pi_{:,1}. \tag{13}$$

The results are displayed in Fig. 10 and Tab. 6. Notably, without Gumbel Softmax, although the model provide slightly better mBO, all the other metrics are kept at the same level as original slot attention model. Moreover, from the visualization, without Gumbel Softmax we can not achieve adaptive instance-level slot selection but produce segmentation with $K_{max} = 24$ masks. This failure is due to the landscape of the soft mask. Consider the following case:

$$\pi_{1,1} = \pi_{2,1} = \cdots = \pi_{K,1}, \quad \text{and} \quad \pi_{K,1} \to 0. \tag{14}$$

Table 5: Ablation study on the designs of masked slot decoder.

| Models | Pair-Counting | | | | Matching | | | Information | |
|---|---|---|---|---|---|---|---|---|---|
| | ARI | P. | R. | $F_1$ | mBO | CorLoc | Purity | AMI | NMI |
| 24 slots | 61.98 | **88.09** | 57.82 | 67.91 | **30.54** | 85.15 | 68.96 | 77.93 | 78.14 |
| Zero Mask | **75.30** | 84.74 | 78.64 | **80.20** | 29.47 | **90.09** | **80.12** | 82.32 | 82.45 |
| Zero Slot† | 0.00 | 21.19 | **100.00** | 33.93 | 2.21 | 0.08 | 33.87 | 0.00 | 0.00 |
| Leanrnable Slot† | 0.00 | 21.19 | **100.00** | 33.93 | 2.21 | 0.08 | 33.87 | 0.00 | 0.00 |

Table 6: Ablation study on the necessity of Gumbel Softmax.

| Models | Pair-Counting | | | | Matching | | | Information | |
|---|---|---|---|---|---|---|---|---|---|
| | ARI | P. | R. | $F_1$ | mBO | CorLoc | Purity | AMI | NMI |
| 24 slot | *61.98* | **88.09** | 57.82 | *67.91* | *30.54* | 85.15 | *68.96* | *77.93* | *78.14* |
| With Gumbel | **75.30** | 84.74 | **78.64** | **80.20** | 29.47 | **90.09** | **80.12** | **82.32** | **82.45** |
| wo Gumbel | 61.76 | *87.49* | *57.88* | 67.74 | **31.31** | *88.85* | 68.85 | 77.51 | 77.73 |

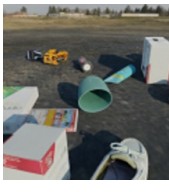 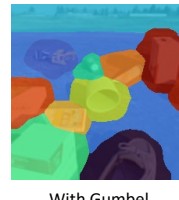 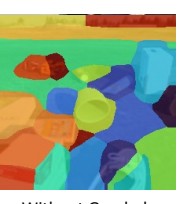 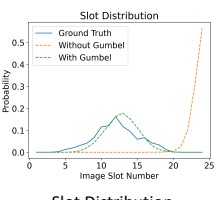

| Raw Image | With Gumbel | Without Gumbel | Slot Distribution |
|---|---|---|---|

Figure 10: Illustration of the segmentation mask without Gumbel softmax and with Gumbel softmax respectively.

The regularization term approach zero $\mathcal{L}_{reg} \to 0$, and $\tilde{m}_i \approx m_i$. Therefore, our method is reduced to ordinary slot attention reconstruction. This simple case shows that without Gumbel Softmax, we can not easily suppress the information of unselected slots, leading to the failure of slot selection. With Gumbel Softmax, when $\pi_{i,1} \to 0$, $Z_i = 0$ and $\tilde{m}_i = 0$ happens with higher probability. The information of $S_i$ will be totally removed. This difference leads to our success.

**Influence of $\lambda$.** We test how the regularization strength $\lambda$ influences the results on MOVi-E. We compare 7 possible values of $\lambda$, ranging from $1e - 2$ to 1 in Tab. 7, keeping other parameters unchanged compared with the main experiments. Generally, larger regularization prefers fewer slots left and grouping more patches. Recall and $\lambda$ exhibit a positive correlation, while Precision and $\lambda$ exhibit a negative correlation. For foreground grouping, the two metrics reach the balance around $\lambda = 0.1$ and $\lambda = 0.2$, which leads to the highest ARI and $F_1$ score. The grouping results have the best agreement with ground truth, which can also be proven by the highest AMI, NMI and Purity score. However, if we continue increasing $\lambda$, these metrics will decrease and drop to an abysmal level. When $\lambda = 1$, the model simply merges all tokens into a single group, which leads to perfect Recall but inferior results for all other metrics. For localization, $\lambda = 0.1$ have the best CorLoc score and performs well on mBO.

## 6.6 RESULTS OF SEMANTIC-LEVEL MASKS ON COCO

In the main paper, we evaluate the metrics on COCO according to the instance mask. Moreover, we report the results based on semantic-level masks in Tab. 8 for further understanding. Compared with instance-level results, grouping metrics like ARI and $F_1$ score are lower, indicating that the

Table 7: Ablation on the influence of different $\lambda$

| $\lambda$ | Pair-Counting | | | | Matching | | | Information | |
|---|---|---|---|---|---|---|---|---|---|
| | ARI | P. | R. | $F_1$ | mBO | CorLoc | Purity | AMI | NMI |
| 0.01 | 62.99 | *87.44* | 59.50 | 68.93 | **30.47** | 85.54 | 69.84 | 78.05 | 78.26 |
| 0.02 | 63.68 | **87.49** | 60.36 | 69.55 | *30.16* | 85.08 | 70.48 | 78.35 | 78.55 |
| 0.05 | 70.95 | 85.67 | 71.48 | 76.32 | 29.46 | *86.79* | 76.67 | 80.87 | 81.02 |
| 0.1 | *75.30* | 84.74 | 78.64 | *80.20* | 29.47 | **90.09** | *80.12* | **82.32** | **82.45** |
| 0.2 | **76.07** | 78.79 | 86.51 | **81.30** | 26.28 | 86.68 | **80.49** | *81.50* | *81.61* |
| 0.5 | 33.52 | 38.04 | *89.27* | 51.74 | 9.05 | 13.26 | 50.40 | 46.62 | 46.74 |
| 1.00 | 0.01 | 21.20 | **99.96** | 33.93 | 2.21 | 0.08 | 33.87 | 0.03 | 0.03 |

Table 8: Experiments of Semantic-level masks on COCO datasets.

| Model | $K$ | Pair-Counting | | | | Matching | | | Information | |
|---|---|---|---|---|---|---|---|---|---|---|
| | | ARI | P. | R. | $F_1$ | mBO | CorLoc | Purity | AMI | NMI |
| DINOSAUR | 4 | 20.72 | 85.02 | 52.18 | 61.47 | 20.61 | 13.89 | 59.32 | 24.93 | 24.96 |
| | 6 | 28.93 | 89.92 | 58.04 | 67.12 | 30.85 | 41.00 | 65.43 | 32.35 | 32.38 |
| | 7 | 27.43 | 90.66 | 54.15 | 64.17 | 31.10 | 39.79 | 62.48 | 31.72 | 31.75 |
| | 8 | 25.32 | 91.29 | 48.89 | 59.89 | 29.93 | 34.69 | 58.31 | 30.74 | 30.78 |
| | 10 | 23.02 | 92.16 | 43.58 | 55.26 | 29.75 | 32.47 | 53.84 | 29.71 | 29.75 |
| | 12 | 20.99 | 92.89 | 38.80 | 50.79 | 29.17 | 30.70 | 49.68 | 28.75 | 28.79 |
| | 20 | 15.72 | 94.12 | 27.97 | 39.43 | 26.44 | 23.33 | 39.35 | 25.77 | 25.82 |
| | 33 | 11.60 | 95.07 | 19.99 | 30.04 | 24.03 | 18.87 | 30.97 | 23.14 | 23.21 |
| Ours | | 26.60 | 89.71 | 55.30 | 64.90 | 30.53 | 37.74 | 63.62 | 30.56 | 30.59 |

Table 9: Comparsion between our model and MaskCut.

| Model | Dataset | Pair-Counting | | | | Matching | | | Information | |
|---|---|---|---|---|---|---|---|---|---|---|
| | | ARI | P. | R. | $F_1$ | mBO | CorLoc | Purity | AMI | NMI |
| MaskCut | MOVi-E | 54.14 | 55.59 | 86.49 | 65.48 | 25.28 | 92.80 | 65.56 | 63.87 | 63.99 |
| | MOVi-C | 59.05 | 75.60 | 88.08 | 79.19 | 40.84 | 88.71 | 78.36 | 60.80 | 60.88 |
| | COCO | 29.18 | 73.58 | 74.47 | 69.73 | 33.95 | 71.88 | 69.23 | 32.20 | 32.25 |
| Ours | MOVi-E | 77.48 | 86.18 | 80.19 | 81.83 | 30.43 | 93.20 | 81.67 | 84.08 | 84.19 |
| | MOVi-C | 72.81 | 86.13 | 86.08 | 84.33 | 37.33 | 80.16 | 83.81 | 75.97 | 76.03 |
| | COCO | 40.38 | 81.26 | 67.16 | 68.55 | 26.94 | 47.12 | 67.33 | 45.53 | 45.59 |

model prefers instance-level object discovery to class-level. Overall, the results of semantic-level and instance-level masks are consistent.

## 6.7 COMPARISON WITH UNSUPERVISED MULTIPLE INSTANCE SEGMENTATION METHOD

Our work falls in unsupervised object discovery, which aims to locate and distinguish between different objects in the image without supervision. However, it does not necessarily provide fine-grained segmentation of each object. In different granularity, unsupervised instance segmentation aims to get a detailed mask for each localized object, clearly demarcating its boundaries.

Most unsupervised object segmentation methods follow a pipeline: initially creating pseudo masks using a self-supervised backbone and subsequently training a segmentation model based on these pseudo masks. In our discussion, we will primarily concentrate on the *initial stage* of these models. We compare our model with MaskCut proposed in CutLER (Wang et al., 2023), since it can generate multiple instance segmentation while other methods either segment only one object from each image (Caron et al., 2021; Wang et al., 2022b), or generate overlapping masks (Wang et al., 2022a). However, it's worth mentioning that MaskCut's inference speed is notably slow, so we work with a fixed subset here.

Table. 9 demonstrate that our model is great at distinguishing objects apart, whereas MaskCut is good at creating masks that closely match objects (thought some masks might cover more than one object). Unlike our model, MaskCut is based on iterative application of Normalized Cuts, which assumes images have very clear foreground and background distinctions, with only a few objects standing out in the foreground. But this assumption does not hold true for MOVi-E/C datasets. As a result, MaskCut produces high-quality masks that capture object shapes well (higher mBO on MOVi-C and COCO), but it struggles to tell different objects apart (lower ARI). This happens because it often groups multiple objects as foreground in each iteration of Normalized Cuts.

*Additionally*, MaskCut takes around several seconds to handle one image, while our model can do object grouping in real-time.

Table 10: Experiments of object property prediction

| Slot | Accuracy | $R^2$ | Accuracy with Correction |
|------|----------|-------|--------------------------|
| 6 | 89.55 | 0.57 | 61.39 |
| 9 | 89.44 | 0.60 | 58.61 |
| 11 | 91.42 | 0.68 | 51.71 |
| Ours | 94.44 | 0.75 | 86.12 |

## 6.8 RESULTS ON OBJECT PROPERTY PREDICTION ON CLEVR10

Except for object discovery, we study the usefulness of the adaptive slot attention for downstream task by object property predicion task. For simplicity, we predict object properties (specifically object position and color) using the CLEVR10 dataset, since the complexity of objects in MOVi-C/E poses challenges in establishing a definition for object property like color.

Our experiments employ a one-hidden layer MLP as the downstream model. The model operates independently on the retained slots. Specifically, a kept slot serves as the model's input, yielding a vector containing property predictions for that particular slot. Our loss function comprises the summation of cross-entropy loss for color classification and MSE loss or coordinate predictions. We align predictions with targets with the Hungarian algorithm Kuhn (1955), minimizing the total loss of the assignment. Following Dittadi et al. (2021), we present results in terms of color classification accuracy and the regression $R^2$ score of position estimation.

To better consider classification correctness and slot count simultaneously, we introduced a classification accuracy measure with a correction factor. For a given image, let $p$, $q$, $m$, and $n$ respectively represent the number of correctly-classified matched predictions, total matched predictions/objects, unmatched predictions, and unmatched ground truth objects. We define:

$$\text{Accuracy with Correction} = \frac{p}{q + m + n},$$

where $m, n$ in the denominator penalizes duplicate prediction and insufficient prediction problems. Our model demonstrates superior performance in terms of accuracy and $R^2$ score, highlighting the effectiveness of our adaptive slot mechanism in enhancing feature quality and boosting downstream tasks. Noting that Accuracy with Correction diminishes alongside the growth of $K$ for the fixed slot model. This observation implies that the fixed large slot model generates redundant slots. Notably, our proposed model achieves the highest Accuracy w/wo Correction, affirming the efficacy of the adaptive slot mechanism we introduced.

## 6.9 MORE VISUALIZATION

To provide a more comprehensive understanding of our methods, we have included additional visualizations in Fig. 11, Fig. 12 and Fig. 13. For each dataset, we select five examples and compare our model with GENESIS-V2 and fixed-slot DINOSAUR. Our model segments the raw image into regions that are not only semantically coherent but also highly meaningful. Moreover, our model showcases adaptability by dynamically adjusting the slot number in accordance with the complexity of the images. It is worth noting that GENESIS-V2 can produce predictions with smaller slot number than the predefined maximum slot number, or generate masks with negligible area for human. For example, in the second last row of Fig. 11, only 8 masks can be perceived by the naked eye.

## 6.10 DISCUSSION AND LIMITATIONS

Our model primarily applies to cases with clearly defined and thoroughly segmented objects. For situations similar to COCO, with numerous complex objects and incomplete annotations, the learned objects may not necessarily align with manual annotations. Additionally, due to the characteristics of the feature reconstruction, the performance on dense small objects is not very outstanding. When compare our model of $K_{max}$ with the fixed slot model of $K = K_{max}$, our model produces fewer masks, and more small objects will be missed. However, the fixed-slot counterpart will also over-segment one object into multiple parts. Further, the part-whole hierarchy in real-world scenes brings

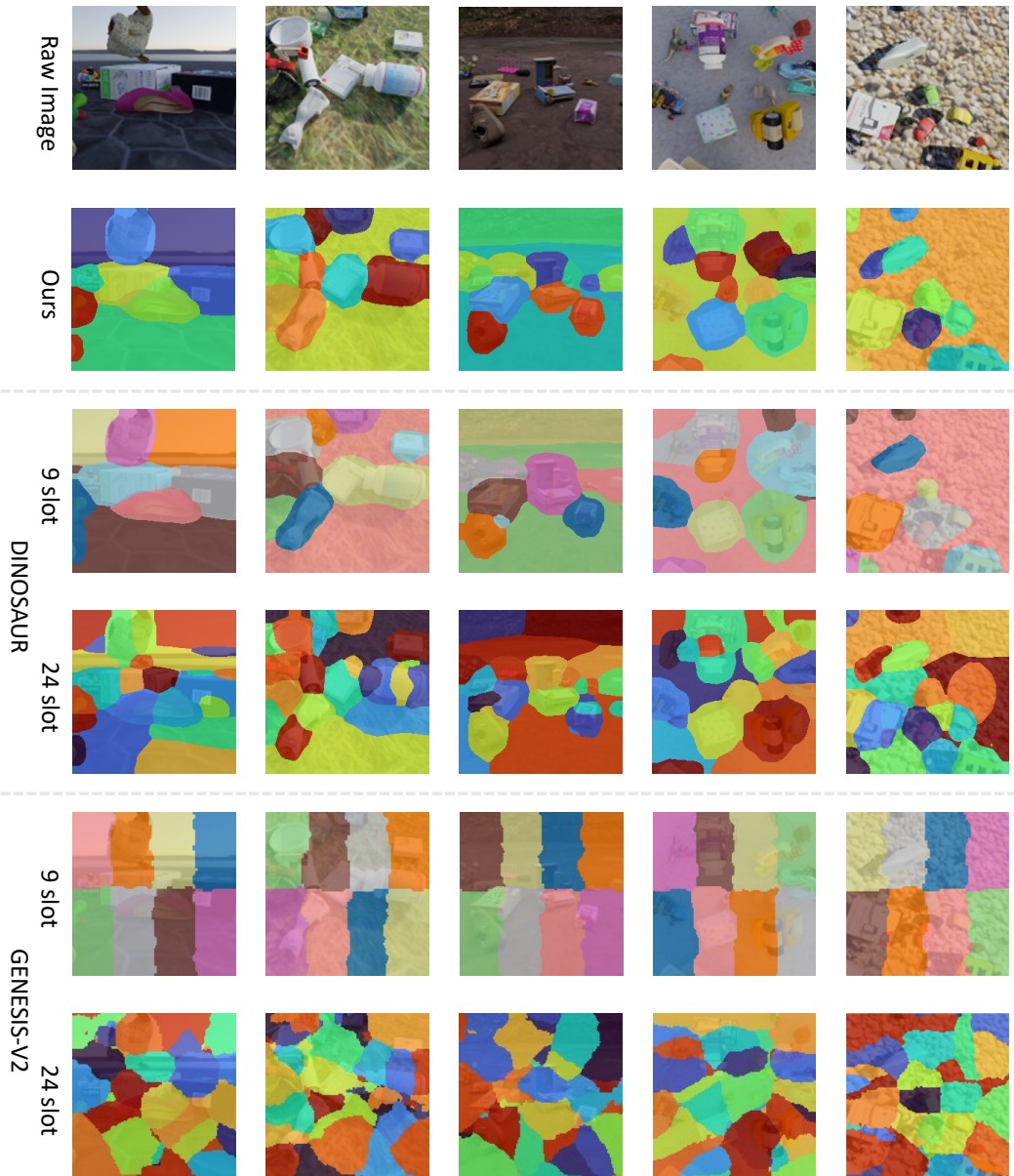

Figure 11: More visualizations on MOVi-E

additional complexity and challenge to unsupervised object discovery. We leave improvements regarding this challenge for future works.

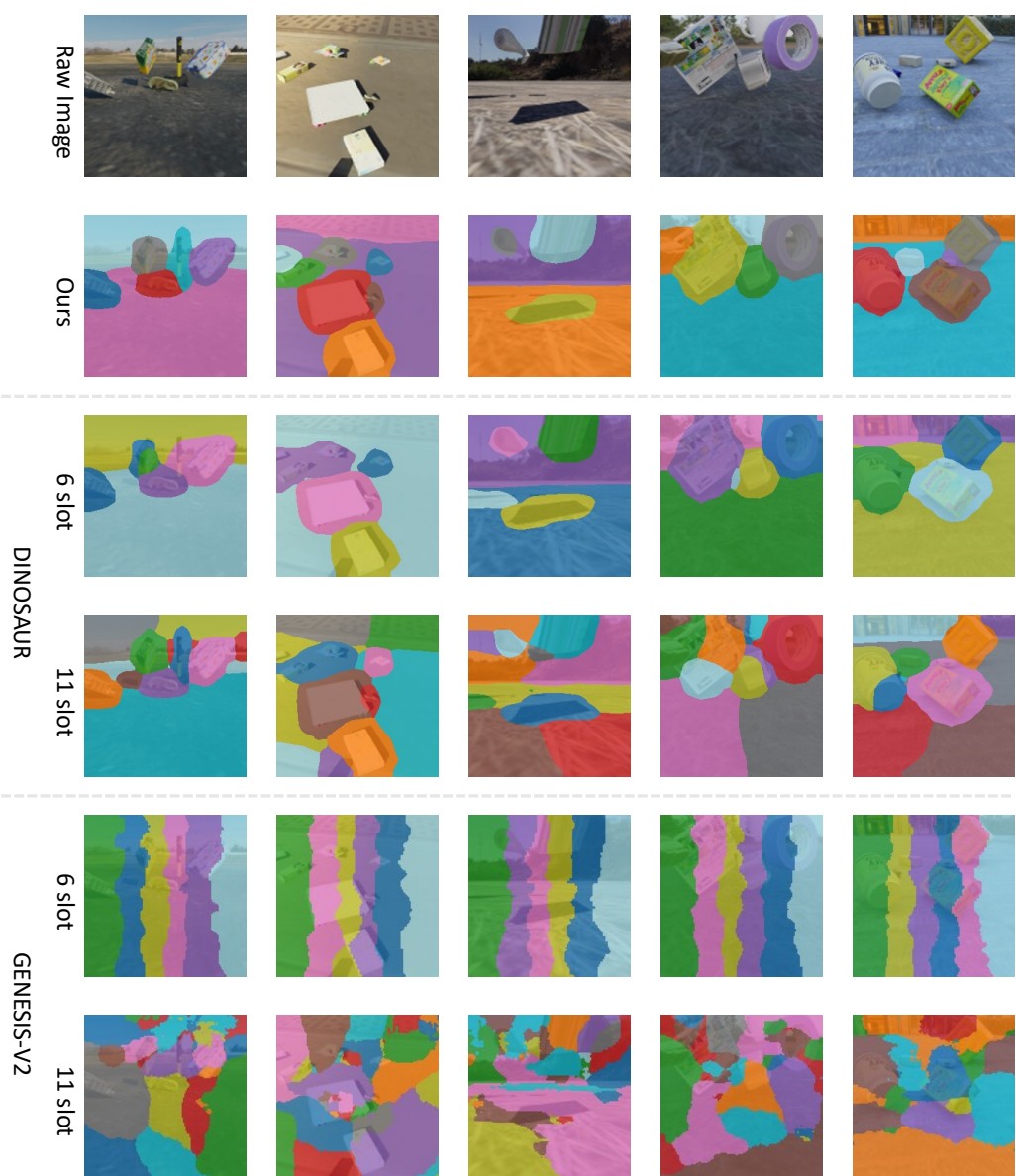

Figure 12: More visualizations on MOVi-C

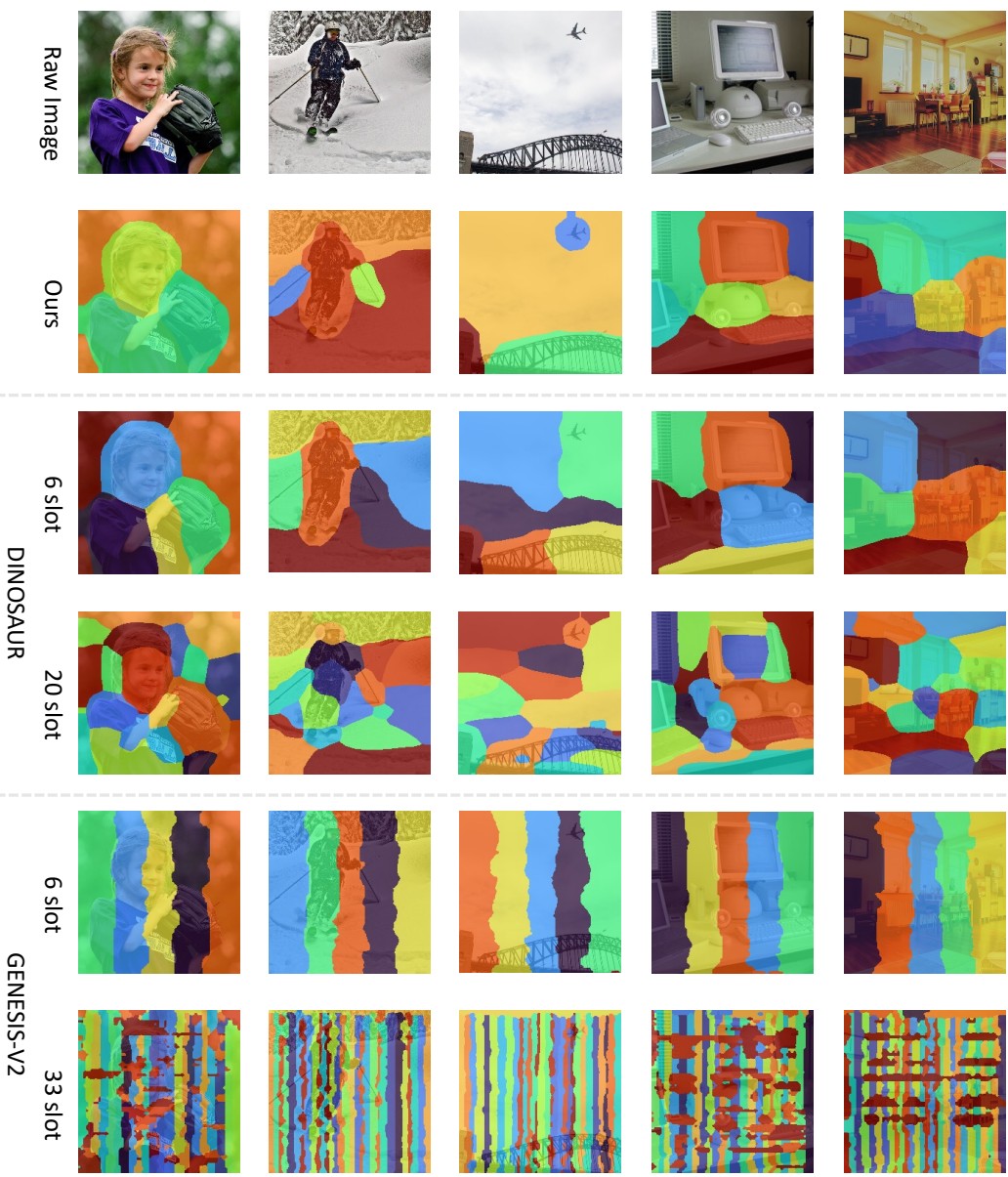

Figure 13: More visualizations on COCO

