# OpenReview forum: "Adaptive Slot Attention: Object Discovery with Dynamic Slot Number"
_ICLR.cc/2024/Conference — ICLR 2024 Conference Withdrawn Submission_

### Official Review · Reviewer_zxD3 · 2023-10-14

**Soundness:** 2 fair
**Presentation:** 3 good
**Contribution:** 2 fair
**Rating:** 5
**Confidence:** 4

**Summary:**

This paper aims to address a persistent challenge in object-centric learning (OCL) -- the need to determine a predefined number of slots. The authors propose to use Gumbel Softmax to dynamically select the number of slots. To train the model, they introduce a masked slot decoder to filter out non-selected slots from the model output. Experimental results on CLEVR10, MOVi-C/E, and COCO demonstrate the effectiveness of this method in adapting the number of slots, leading to consistently better results compared to OCL with fixed number of slots.

**Strengths:**

- The paper is well-written and easy to follow
- The number of slots as a hyper-parameter is indeed a long-standing problem in the field. This paper proposes a valid solution to it
- The experiments show promising segmentation results on both synthetic and real-world datasets

**Weaknesses:**

I have two main concerns regarding the paper:
- The experiments in the paper mainly focus on object segmentation. While it is an important outcome of OCL, the quality of learned object slots is another important aspect. The authors conduct object property prediction experiments on CLEVR10. However, CLEVR10 is too simple. I would suggest the authors to further experiment on MOVi, e.g., follow the protocol in LSD [1]
- The goal of this paper is to get rid of a pre-defined number of slots. However, there is still a hyper-parameter $K_{max}$. I understand that this is necessary for implementation. But $K_{max}$ on CLEVR, MOVi-C/E is actually the maximum number of objects on that dataset plus one. Therefore, this still utilizes the prior knowledge of the dataset. On COCO, $K_{max} = 33$ seems to be an arbitrary number. I wonder how will the model perform if using $K_{max} = 30$ on MOVi-C/E?

[1] Jiang, Jindong, et al. "Object-centric slot diffusion." arXiv preprint arXiv:2303.10834 (2023).

**Questions:**

Besides weaknesses, I have two minor questions:
- Does the GENESIS-V2 baseline use the same DINO ViT encoder? If not, that seems like an unfair comparison. But I somehow feel that is fine since DINOSAUR is the current SOTA OCL method
- Can the masked slot decoder technique be easily extended to the Transformer-based slot decoder, like SLATE and STEVE?

---

> ### Author Response · Authors · 2023-11-15
>
> **Question:** The experiments in the paper mainly focus on …… follow the protocol in LSD [1]
>
> **Answer:** Thanks! We conduct the property prediction on CLEVER10 due to the simplicity of defining the property for the object. For more complex objects, we will try the protocol of  LSD on the dataset MOVi.
>
> **Question:** Hyper-parameter $K_{max}$
>
> **Answer:** Currently, $K_{max}$ is the maximum number of objects on that **dataset** plus one, which is a very weak prior compared with knowing the specific number of objects for each **instance**.  We only need to roughly estimate the object number and set the $K_{max}$.
>
> We acknowledge this might be a potential limitation, which will be discussed in the revised paper. However, we would like to emphasize that the relaxed requirement from instance-level hyper-parameter to dataset-level hyper-parameter plays an important step towards a fully automatic slot number selection.
>
> **Question:** Can the masked slot decoder technique be easily extended to the Transformer-based slot decoder, like SLATE and STEVE?
>
> **Answer:** Sure, this is an open question of exploring dynamic slots, particularly in transformer-based decoders. Currently, our experiments use the most classical ones -- mixture decoders; and our results are solid enough to support our contributions by dynamically varying the number of slots.
>
>
> A potential generalization is that we could mask the elements of the attention matrix related to the dropped slot in the transformer decoder. That may demand extensive exploration to check whether the technique mentioned in the rebuttal will work or not on transformer-based decoders. We consider this as a potential future work.

---

### Official Review · Reviewer_BxGH · 2023-10-26

**Soundness:** 2 fair
**Presentation:** 3 good
**Contribution:** 2 fair
**Rating:** 5
**Confidence:** 5

**Summary:**

The authors propose a method for dynamically adjusting the number of slots for slot attention so that it can vary across inputs. Each slot predicts whether it "wants" to be kept or not, from which a mask is sampled using Gumbel Softmax. If a slot is dropped, its influence on the attention is set to 0. There is an additional regularization term that encourages lower numbers of slots.

**Strengths:**

- Reasonable approach to the problem while still being computationally efficient and differentiable
- Considers a problem that seems useful, improving the determination of the "correct" number of objects
- Extensive ablations and analysis to understand the method are impressive

**Weaknesses:**

- The premise of the paper hinges on the assumption that it is desirable to have the slots correspond exactly to objects. While this appears reasonable in simpler datasets, the notion of what an object is becomes less clear for more realistic datasets (such as COCO, which only has certain types of objects labeled at a very particular granularity level). In the end, the goal of unsupervised object discovery is to do something useful with the objects. Therefore, I believe that when arguing whether one decomposition over slots (normal slot attention) is better or worse than another (the variant proposed in this paper), it should be tied to downstream performance, less to accuracy of the provided "ground-truth" masks; especially in the case of natural images, where objects of different levels of abstraction are valid at the same time. My worry is therefore that this method works well on simple images where there is an obvious notion of object, but does not scale well to the case of realistic images. I believe it would be highly beneficial if the proposed model can be evaluated on *downstream performance* on something that is more realistic, which (given the use of DINOSAUR already) should not be unsurmountable. So far, I only see the downstream performance for CLEVR10, which is a simple dataset (with admittedly better performance for the proposed approach). This is my main concern, I am willing to change my score if this is sufficiently addressed.
- There are no error bars on the experiments

**Questions:**

- It would be interesting to see an oracle baseline where you use DINOSAUR, but provide the ground-truth number of slots per instance and compare performance. Does your model approach the performance of this (unrealistic) model that knows the ground-truth number of objects?
- Can the maximum number of slots for your model K_max be set arbitrarily high? Are there performance trade-offs for that?
- I did not understand the second part of the sentence: "We set λ to 0.1 for MOVi-E/C and 0.5 for COCO, without specifying a particular claim". What does specifying a particular claim mean here?
- What does "flattening the videos" mean in the context of turning the MOVi video dataset into an image dataset?
- Figure 7, how is the prediction for the number of objects trained and made?

---

> ### Author Response · Authors · 2023-11-15
>
> **Question 1:** The premise …… if this is sufficiently addressed.
>
> **Answer:** Thanks for your valuable suggestion. We acknowledge that the complexity of real-world data makes the definition of optimal objects/slots number a challenging (and somehow ambiguous) problem. We will try our best to conduct experiments on real-world dataset with evaluation on more downstream performance. It would be great if you could provide more information on the downstream performance you mentioned.
>
> **Question 2:** no error bars on the experiments
>
> **Answer:** Thanks! We may revisit the problem to better measure the uncertainty of our algorithms.
>
> **Question 3:** It would be interesting to see an oracle baseline where you use DINOSAUR, but provide the ground-truth number of slots per instance and compare performance. Does your model approach the performance of this (unrealistic) model that knows the ground-truth number of objects?
>
> **Answer:** Thanks for your valuable suggestion! We are not sure if the time of the rebuttal period is enough to achieve this result. We acknowledge this experiment would provide more insight. We will try our best to make it and include the result as well as the discussion in the revised version.
>
> **Question 4:** Can the maximum number of slots for your model $K_{max}$ be set arbitrarily high? Are there performance trade-offs for that?
>
> **Answer:** Generally, the larger $K$($K_{max}$) is, the slot attention mechanism tends to produce more segmentations.  The performance will be good when the produced slot number roughly matches the dataset. So $K_{max}$ can not be set arbitrarily high. When $K_{max}$ is set larger, we can slightly increase $\lambda$ to strengthen the regularization to control the expectation of slots produced by the adaptive slot mechanism.
>
> **Question 5:** I did not understand the second part of the sentence: "We set λ to 0.1 for MOVi-E/C and 0.5 for COCO, without specifying a particular claim". What does specifying a particular claim mean here?
>
> **Answer:** In the appendix, we add more ablation studies, especially varying the  $\lambda$ to study its impact. We mean all the experiments other than the ablation study of varying  $\lambda$ utilize this hyperparameter setting.
>
> **Question 6:** What does "flattening the videos" mean in the context of turning the MOVi video dataset into an image dataset?
>
> **Answer:** It means that we treat the videos as a collection of images without using the sequence information and treat MOVi as an image dataset. We will make it clearer in the revised version.
>
> **Question 7:** Figure 7, how is the prediction for the number of objects trained and made?
>
> **Answer:** In Figure 7, we set the $K_{max}$ to 11 on CLEVER10 and train the adaptive slot mechanism since CLEVER10 contains 10 objects at most. For each image, we record the GT number of objects and the predicted number of slots by our model and make the confusion matrix and visualize it.

---

### Official Review · Reviewer_pNUM · 2023-11-01

**Soundness:** 3 good
**Presentation:** 3 good
**Contribution:** 2 fair
**Rating:** 5
**Confidence:** 3

**Summary:**

This paper introduces an adaptive slot attention mechanism for the problem of object discovery. On observing that current object-centric approaches, such as slot attention, rely on predefining the number of slots, they propose a complexity-aware object auto-encoder framework to dynamically determine the optimal number of slots based on the content of the data. Extensive experiments on standard benchmarks reveal the effectiveness of the proposed framework in terms of object discovery tasks.

**Strengths:**

- The proposed method is well-motivated.

- The proposed method is shown to be effective in dynamically the number of slots within the attention mechanism.

- The proposed method is simple and easy to implement.

**Weaknesses:**

- The proposed method has little novelty since it is a straightforward combination of existing methods: slot attention (Locatello et al., 2020), clustering number selection (Blei & Jordan, 2006), and Gumbel-Softmax (Jang et al., 2016). The authors are encouraged to offer more insights into how these modules interact with each other and the specific roles they play individually. This added detail would contribute to a deeper understanding of the method's functionality and its overall design rationale.

- The experimental scope of the paper is somewhat limited, with the authors only utilizing two real-world datasets. I would recommend that they consider expanding their experiments to include additional datasets, such as commonly used segmentation datasets, to provide a more comprehensive evaluation of their method and strengthen the validity of their results.

- The logical flow of the manuscript requires improvement as there are numerous sentences throughout the paper that lack clarity and coherence. Attention to sentence structure and overall organization is necessary to enhance readability and ensure that the authors' arguments are conveyed effectively.

**Questions:**

Please refer to the weaknesses.

---

> ### Author Response · Authors · 2023-11-15
>
> **Question 1:** The proposed method has ……  and its overall design rationale.
>
> **Answer:**  We aim to improve the slot attention such that it can adaptively choose the slot number according to specific instances. To achieve this, we formulate the problem as a subset selection problem and utilize the Gumbel-Softmax to make the selection differentiable. Moreover, we follow Blei & Jordan and utilize the mean-field formulation to improve the efficiency of Gumbel-Softmax subset selection. The full approach is designed in a problem-driven way step by step.
>
> **Question 2:** The experimental scope of the paper…… strengthen the validity of their results.
>
> **Answer:** In the literature of object centric learning, most methods focus on toy synthesis datasets such as CATER and CLEVER. Besides CLEVER10, we carry out experiments on MOVi-C/E, a more complex dataset with high-resolution scanned objects, and COCO, a real-world dataset. Our experiment is solid enough to support our contributions of dynamically varying the number of slots.
>
> **Question 3:** The logical flow of the manuscript requires improvement as there are numerous sentences throughout the paper that lack clarity and coherence. Attention to sentence structure and overall organization is necessary to enhance readability and ensure that the authors' arguments are conveyed effectively.
>
> **Answer:**  Thanks for your suggestion! We will improve our writing in the revision of our manuscripts.

---

### Official Review · Reviewer_CDqw · 2023-11-06

**Soundness:** 2 fair
**Presentation:** 2 fair
**Contribution:** 2 fair
**Rating:** 5
**Confidence:** 3

**Summary:**

This paper proposes a novel method called adaptive slot attention, which could select an appropriate number of slots automatically based on the complexity of the instance, better to serve slot attention for object-centric learning tasks. In particular, a discrete slot sampling module and a masked slot decoder are designed to work for the adaptive mechanism, and the experiments conducted on benchmark datasets verify the stability and effectiveness of the proposed method.

**Strengths:**

1. To design a module to adaptively select the number of slots used for slot attention is an interesting and exciting research direction.

2. The complexity-aware object-encoder framework is designed simple and clear. It makes the readers easily understand the function of every single component.

3. This paper provides a variety of experimental studies to better demonstrate the properties of the proposed method.

**Weaknesses:**

1. This paper is a bit difficult to follow. For example, this paper proposes an object-centric learning method, but the title is about one of its applications, object discovery. Meanwhile, there are no experiments about object discovery to further support the title. It makes the readers confused about the target of this paper. In addition, the writing of the method part could be better improved. For example, it is difficult to follow the authors to understand what are the structures of the two proposed modules. It could be better to provide a framework or pipeline figure to demonstrate the whole structure. Also, for the equations, there are several important notion descriptions missing and ambiguous. For example, in the preliminary, is F \in \mathcal R^{H \times W} or F \in \mathcal R^{C \times H \times W}. The encoded features should have a dimension if I'm correct. Meanwhile, what is the notion \alpha_i? What is the operation between m_i and x_i in the first term of Eq(2)? Moreover, how the original slot attention works could be better explained in this part. In the second paragraph on page 4, what is the relationship between the instance \mathbf x and the image x_i that appeared in Eq(2)? This part could easily make the readers confused. For the sentence after Eq(5), where are \pi_{i,0} and \pi_{i,1}? They are not mentioned in the previous part but suddenly appear in this explanation.

2. It could be better to rearrange the experimental part. Firstly, since the paper aims at designing a slot number selection mechanism, to provide a comparison between ground truth on the number of the objects in the image and the predicted number generated by the proposed method. If the proposed method could correctly predict the number of objects within an image according to the designed complexity-aware object-encoder, then it will give more convincing evidence to support the contribution claimed by this paper. In addition, since the previous methods have to pre-define the number of slots, a plot built on the evaluation metric vs the number of slots will better demonstrate the comparison between these methods and the proposed one. In addition, such a plot could better show the stability of this work.

**Questions:**

1. Please give a better description of the slot attention and the whole framework of the proposed method.
2. Please give more detail about the number of slots predicted by the proposed method and give a comparison between the ground truth and the predicted ones.

**Details Of Ethics Concerns:**

Nil

---

> ### Author Response · Authors · 2023-11-15
>
> **Weaknesses 1:**  This paper is a bit difficult to follow ……They are not mentioned in the previous part but suddenly appear in this explanation.
>
> **Answer:** Thanks for the suggestion! Object-discovery is one of the most straightforward tasks to validate object-centric learning methods, thus we put our efforts on this part. The experiments of MOVI-C/E and COCO dataset is about the object discovery task and we provide the metrics such as FG-ARI. We will improve our writing to make this part clearer.
>
> **Weaknesses 2:** It could be better to rearrange the experimental part …… In addition, such a plot could better show the stability of this work.
>
> **Answer:** Thanks for your advice. We will improve the experimental part in the revision of our manuscripts.
>
> **Question 1:** Please give a better description of the slot attention and the whole framework of the proposed method.
>
> **Answer:** Thanks. We will consider providing an overview figure of the full pipeline to demonstrate the architecture and slot attention mechanism to make it easier for readers to understand.
>
> **Question 2:** more detail about the number of slots predicted by the proposed method and give a comparison between the ground truth and the predicted ones.
>
> **Answer:** We have described and displayed the comparison of the ground truth and the predicted slots in the paper, please refer to the Figure 4 and 7 and section 4.2 and Analyzing the Slot Selection Process.